# BlockGAN: Learning 3D Object-aware Scene Representations from Unlabelled Images

**Thu Nguyen-Phuoc**
University of Bath

**Christian Richardt**
University of Bath

**Long Mai**
Adobe Research

**Yong-Liang Yang**
University of Bath

**Niloy Mitra**
Adobe Research & UCL

## Abstract

We present BlockGAN, an image generative model that learns object-aware 3D scene representations directly from unlabelled 2D images. Current work on scene representation learning either ignores scene background or treats the whole scene as one object. Meanwhile, work that considers scene compositionality treats scene objects only as image patches or 2D layers with alpha maps. Inspired by the computer graphics pipeline, we design BlockGAN to learn to first generate 3D features of background and foreground objects, then combine them into 3D features for the whole scene, and finally render them into realistic images. This allows BlockGAN to reason over occlusion and interaction between objects' appearance, such as shadow and lighting, and provides control over each object's 3D pose and identity, while maintaining image realism. BlockGAN is trained end-to-end, using only unlabelled single images, without the need for 3D geometry, pose labels, object masks, or multiple views of the same scene. Our experiments show that using explicit 3D features to represent objects allows BlockGAN to learn disentangled representations both in terms of objects (foreground and background) and their properties (pose and identity). Our code is available at https://github.com/thunguyenphuoc/BlockGAN.

## 1   Introduction

The computer graphics pipeline has achieved impressive results in generating high-quality images, while offering users a great level of freedom and controllability over the generated images. This has many applications in creating and editing content for the creative industries, such as films, games, scientific visualisation, and more recently, in generating training data for computer vision tasks. However, the current pipeline, ranging from generating 3D geometry and textures, rendering, compositing and image post-processing, can be very expensive in terms of labour, time, and costs.

Recent image generative models, in particular generative adversarial networks [GANs; 14], have greatly improved the visual fidelity and resolution of generated images [5, 23, 24]. Conditional GANs [36] allow users to manipulate images, but require labels during training. Recent work on unsupervised disentangled representations using GANs [9, 24, 38] relaxes this need for labels. The ability to produce high-quality, controllable images has made GANs an increasingly attractive alternative to the traditional graphics pipeline for content generation. However, most work focuses on *property* disentanglement, such as shape, pose and appearance, without considering the compositionality of the images, i.e., scenes being made up of multiple objects. Therefore, they do not offer control over individual objects in a way that respects the interaction of objects, such as consistent lighting and shadows. This is a major limitation of current image generative models, compared to the graphics pipeline, where 3D objects are modelled individually in terms of geometry and appearance, and combined into 3D scenes with consistent lighting.

Even when considering object compositionality, most approaches treat objects as 2D layers combined using alpha compositing [12, 50, 53]. Moreover, they also assume that each object's appearance is independent [3, 6, 12]. While this layering approach has led to good results in terms of object separation and visual fidelity, it is fundamentally limited by the choice of 2D representation. Firstly, it is hard to manipulate properties that require 3D understanding, such as pose or perspective. Secondly, object layers tend to bake in appearance and cannot adequately represent view-specific appearance, such as shadows or material highlights changing as objects move around in the scene. Finally, it is non-trivial to model the appearance interactions between objects, such as scene lighting that affects objects' shadows on a background.

We introduce BlockGAN, a generative adversarial network that learns 3D object-oriented scene representations directly from unlabelled 2D images. Instead of learning 2D layers of objects and combining them with alpha compositing, BlockGAN learns to generate 3D object features and to combine them into deep 3D scene features that are projected and rendered as 2D images. This process closely resembles the computer graphics pipeline where scenes are modelled in 3D, enabling reasoning over occlusion and interaction between object appearance, such as shadows or highlights. During test time, each object's pose can be manipulated using 3D transforms directly applied to the object's deep 3D features. We can also add new objects and remove existing objects in the generated image by changing the number of 3D object features in the 3D scene features at inference time. This shows that BlockGAN has learnt a non-trivial representation of objects and their interaction, instead of merely memorizing images.

BlockGAN is trained end-to-end in an unsupervised manner directly from unlabelled 2D images, without any multi-view images, paired images, pose labels, or 3D shapes. We experiment with BlockGAN on a variety of synthetic and natural image datasets. In summary, our main contributions are:

- BlockGAN, an unsupervised image generative model that learns an object-aware 3D scene representation directly from unlabelled 2D images, disentangling both between objects and individual object properties (pose and identity);
- showing that BlockGAN can learn to separate objects even from cluttered backgrounds; and
- demonstrating that BlockGAN's object features can be added, removed and manipulated to create novel scenes that are not observed during training.

## 2   Related work

**GANs.**   Unsupervised GANs learn to map samples from a latent distribution to data categorised as real by a discriminator network. Conditional GANs enable control over the generated image content, but require labels during training. Recent work on unsupervised disentangled representation learning using GANs provides controllability over the final images without the need for labels. Loss functions can be designed to maximize mutual information between generated images and latent variables [9, 20]. However, these models do not guarantee which factors can be learnt, and have limited success when applied to natural images. Network architectures can play a vital role in both improving training stability [7] and controllability of generated images [24, 38]. We also focus on designing an appropriate architecture to learn object-level disentangled representations. We show that injecting inductive biases about how the 3D world is composed of 3D objects enables BlockGAN to learn 3D object-aware scene representations directly from 2D images, thus providing control over both 3D pose and appearance of individual objects.

**3D-aware neural image synthesis.**   Introducing 3D structures into neural networks can improve the quality [37, 41, 44, 48] and controllability of the image generation process [38, 39, 59]. This can be achieved with explicit 3D representations, like appearance flow [58], occupancy voxel grids [43, 59], meshes, or shape templates [27, 46, 56], in conjunction with handcrafted differentiable renderers [8, 17, 31, 33]. Renderable deep 3D representations can also be learnt directly from images [38, 47, 48]. HoloGAN [38] further shows that adding inductive biases about the 3D structure of the world enables unsupervised disentangled feature learning between shape, appearance and pose. However, these learnt representations are either object-centric (i.e., no background), or treat the whole scene as one object. Thus, they do not consider scene compositionality, i.e., components that can move independently. BlockGAN, in contrast, is designed to learn object-aware 3D representations that are combined into a unified 3D scene representation.

**Object-aware image synthesis.**   Recent methods decompose image synthesis into generating components like layers or image patches, and combining them into the final image [28, 50, 53]. This includes

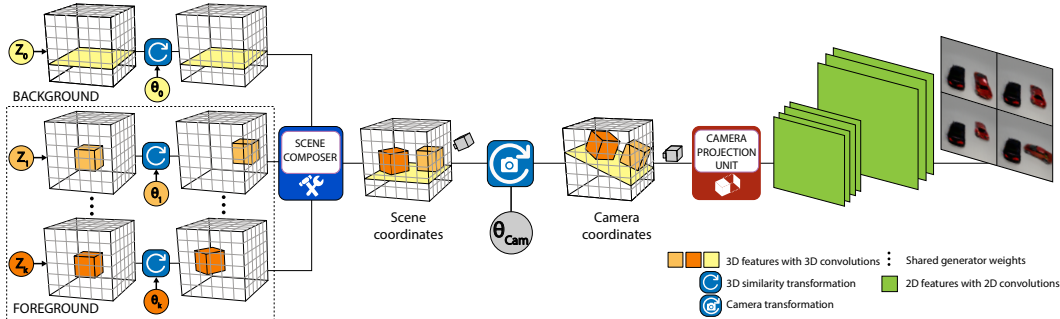

Figure 1: BlockGAN's generator network. Each noise vector $\mathbf{z}_i$ is mapped to deep 3D object features, which are transformed to the desired 3D pose $\boldsymbol{\theta}_i$. Object features are combined into 3D scene features, where the camera pose $\boldsymbol{\theta}_{\text{cam}}$ is applied, before projection to 2D features that produce the image $\mathbf{x}$.

conditional GANs that use segmentation masks [40, 49], scene graphs [22], object labels, key points or bounding boxes [18, 42], which have shown impressive results for natural image datasets. Recently, unsupervised methods [2, 12, 13, 26, 50, 55] learned object disentanglement for multi-object scenes on simpler synthetic datasets (single-colour objects, simple lighting, and material). Other approaches successfully separate foreground from background objects in natural images, but make strong assumptions about the size of objects [53] or independent object appearance [3, 6]. These methods treat object components as image patches or 2D layers with corresponding masks, which are combined via alpha compositing at the pixel level to generate the final stage. The work closest to ours learns to generate multiple 3D primitives (cuboids, spheres and point clouds), renders them into *separate* 2D layers with a handcrafted differentiable renderer, and alpha-composes them based on their depth ordering to create the final image [29]. Despite the explicit 3D geometry, this method does not handle cluttered backgrounds and requires extra supervision in the shape of labelled images with and without foreground objects.

BlockGAN takes a different approach. We treat objects as *learnt 3D features* with corresponding 3D poses, and learn to combine them into 3D scene features. Not only does this provide control over 3D pose, but also enables learning of realistic lighting and shadows. Our approach allows adding more objects into the 3D scene features to generate images with multiple objects, which are not observed at training time.

## 3 Method

Inspired by the computer graphics pipeline, we assume that each image $\mathbf{x}$ is a rendered 2D image of a 3D scene composed of $K$ 3D foreground objects $\{O_1,...,O_K\}$ in addition to the background $O_0$:

$$\mathbf{x} = p\big(f(\underbrace{O_0,}_{\text{background}} \underbrace{O_1,...,O_K}_{\text{foreground}})\big), \tag{1}$$

where the function $f$ combines multiple objects into unified scene features that are projected to the image $\mathbf{x}$ by $p$. We assume each object $O_i$ is defined in a canonical orientation and generated from a noise vector $\mathbf{z}_i$ by a function $g_i$ before being individually posed using parameters $\boldsymbol{\theta}_i$: $O_i = g_i(\mathbf{z}_i, \boldsymbol{\theta}_i)$.

We inject the inductive bias of compositionality of the 3D world into BlockGAN in two ways. (1) The generator is designed to first generate 3D features for each object independently, before transforming and combining them into unified scene features, in which objects interact. (2) Unlike other methods that use 2D image patches or layers to represent objects, BlockGAN directly learns from unlabelled images how to generate objects as 3D features. This allows our model to disentangle the scene into separate 3D objects and allows the generator to reason over 3D space, enabling object pose manipulation and appearance interaction between objects. BlockGAN, therefore, learns to both generate and render the scene features into images that can fool the discriminator.

Figure 1 illustrates the BlockGAN generator architecture. Each noise vector $\mathbf{z}_i$ is mapped to 3D object features $O_i$. Objects are then transformed according to their pose $\boldsymbol{\theta}_i$ using a 3D similarity transform, before being combined into 3D scene features using the *scene composer* $f$. The scene features are transformed into the camera coordinate system before being projected to 2D features to render the final images using the *camera projector* function $p$. During training, we randomly sample both the noise vectors $\mathbf{z}_i$ and poses $\boldsymbol{\theta}_i$. During test time, objects can be generated with a given identity $\mathbf{z}_i$ in the desired pose $\boldsymbol{\theta}_i$.

BlockGAN is trained end-to-end using only unlabelled 2D images, without the need for any labels, such as poses, 3D shapes, multi-view inputs, masks, or geometry priors like shape templates, symmetry or smoothness terms. We next explain each component of the generator in more detail.

**3.1. Learning 3D object representations.** Each object $O_i \in \mathbb{R}^{H_o \times W_o \times D_o \times C_o}$ is a deep 3D feature grid generated by $O_i = g_i(\mathbf{z}_i, \boldsymbol{\theta}_i)$, where $g_i$ is an object generator that takes as input a noise vector $\mathbf{z}_i$ controlling the object appearance, and the object's 3D pose $\boldsymbol{\theta}_i = (s_i, \mathbf{R}_i, \mathbf{t}_i)$, which comprises its uniform scale $s_i \in \mathbb{R}$, rotation $\mathbf{R}_i \in \mathrm{SE}(3)$ and translation $\mathbf{t}_i \in \mathbb{R}^3$. The object generator $g_i$ is specific to each category of objects, and is shared between objects of the same category. We assume that 3D scenes consist of at least two objects: the background $O_0$ and one or more foreground objects $\{O_1, ..., O_K\}$. This is different to object-centric methods that only assume a single object with a simple white background [47], or only deal with static scenes whose object components cannot move independently [38]. We show that, even when BlockGAN is trained with only one foreground and background object, we can add an arbitrary number of foreground objects to the scene at test time.

To generate 3D object features, BlockGAN implements the style-based strategy, which helps to disentangle between pose and identity [38] while improving training stability [24]. As illustrated in Figure 2, the noise vector $\mathbf{z}_i$ is mapped to affine parameters – the "style controller" – for adaptive instance normalization [AdaIN; 19] after each 3D convolution layer. However, unlike HoloGAN [38], which learns 3D features directly for the whole scene, BlockGAN learns 3D features for *each* object, which are then transformed to their target poses using similarity transforms, and combined into 3D *scene* features. We implement these 3D similarity transforms by trilinear resampling of the 3D features according to the translation, rotation and scale parameters $\boldsymbol{\theta}_i$;

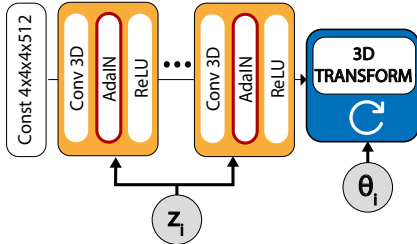

Figure 2: BlockGAN's object generator. Each object starts with a constant tensor that is learnt with the rest of the network.

samples falling outside the feature tensor are clamped to zero. This allows BlockGAN to not only separate object pose from identity, but also to disentangle multiple objects in the same scene.

**3.2. Scene composer function.** We combine the 3D object features $\{O_i\}$ into scene features $S = f(O_0, O_1, ..., O_K) \in \mathbb{R}^{H_s \times W_s \times D_s \times C_s}$ using a scene composer function $f$. For this, we use the element-wise maximum as it achieves the best image quality compared to element-wise summation and a multi-layer perceptron (MLP); please see our supplemental document for an ablation. Additionally, the maximum is invariant to permutation and allows a flexible number of input objects to add new objects into the scene features during test time, even when trained with fewer objects (see Section 4.3).

**3.3. Learning to render.** Instead of using a handcrafted differentiable renderer, we aim to learn rendering directly from unlabelled images. HoloGAN showed that this approach is more expressive as it is capable of handling unlabelled, natural image data. However, their projection model is limited to a weak perspective, which does not support foreshortening – an effect that is observed when objects are close to real (perspective) cameras. We therefore introduce a graphics-based perspective projection function $p \colon \mathbb{R}^{H_s \times W_s \times D_s \times C_s} \mapsto \mathbb{R}^{H_c \times W_c \times C_c}$ that transforms the 3D scene features into camera space using a projective transform, and then learns the projection of the 3D features to a 2D feature map.

The computer graphics pipeline implements perspective projection using a projective transform that converts objects from world coordinates (our scene space) to camera coordinates [34]. We implement this camera transform like the similarity transforms used to manipulate objects in Section 3.1, by resampling the 3D scene features according to the viewing volume (frustum) of the virtual perspective camera (see Figure 3). For correct perspective projection, this transform must be a projective transform, the superset of similarity transforms [52]. Specifically, the viewing frustum, in scene space, can be defined relative to the camera's pose $\boldsymbol{\theta}_{\mathrm{cam}}$ using the angle of view,

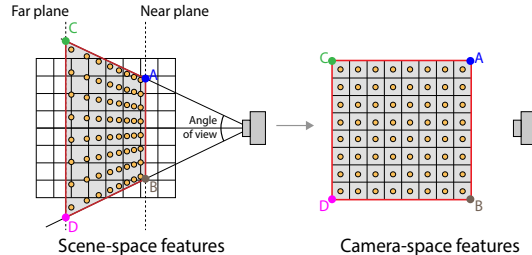

Figure 3: **Left:** The camera's viewing volume (frustum) overlaid on scene-space features. We trilinearly resample the scene features based on the viewing volume at the orange dots. **Right:** The resulting camera-space features before projection to 2D.

and the distance of the near and far planes. The camera-space features are a new 3D tensor of features, of size $H_c \times W_c \times D_c \times C_s$, whose corners are mapped to the corners of the camera's viewing frustum using the unique projective 3D transform computed from the coordinates of corresponding corners using the direct linear transform [16].

In practice, we combine the object and camera transforms into a single transform by multiplying both transform matrices and resampling the object features in a single step, directly from object to camera space. This is computationally more efficient than resampling twice, and advantageous from a sampling theory point of view, as the features are only interpolated once, not twice, and thus less information is lost by the resampling. The combined transform is a fixed, differentiable function with parameters $(\boldsymbol{\theta}_i, \boldsymbol{\theta}_{\text{cam}})$. The individual objects are then combined in camera space before the final projection.

After the camera transform, the 3D features are projected into view-specific 2D feature maps using the *learnt* camera projection $p' : \mathbb{R}^{H_c \times W_c \times D_c \times C_s} \mapsto \mathbb{R}^{H_c \times W_c \times C_c}$. This function ensures that occlusion correctly shows nearby objects in front of distant objects. Following the RenderNet projection unit [37], we reshape the 3D camera-space features (with depth $D_c$ and $C_s$ channels) into a 2D feature map with $(D_c \cdot C_s)$ channels, followed by a per-pixel MLP (i.e., $1 \times 1$ convolution) that outputs $C_c$ channels. We choose to use this learnt renderer following HoloGAN, which shows the effectiveness of the renderer in learning powerful 3D representations directly from unlabelled images. This is different from the supervised multi-view setting with pose labels in the renderer of DeepVoxels [47], which learns occlusion values, or Neural Volumes [32] and NeRF [35], which learn explicit density values.

**3.4. Loss functions.**    We train BlockGAN adversarially using the non-saturating GAN loss [14]. For natural images with cluttered backgrounds, we also add a style discriminator loss [38]. In addition to classifying the images as real or fake, this discriminator also looks at images at the feature level. Given image features $\boldsymbol{\Phi}_l$ at layer $l$, the style discriminator classifies the mean $\boldsymbol{\mu}(\boldsymbol{\Phi}_l)$ and standard deviation $\boldsymbol{\sigma}(\boldsymbol{\Phi}_l)$ over the spatial dimensions, which describe the image "style" [19]. This more powerful discriminator discourages the foreground generator to include parts of the background within the foreground object(s). We provide detailed network and loss definitions in the supplemental material.

# 4 Experiments

**Datasets.**    We train BlockGAN on images at $64 \times 64$ pixels, with increasing complexity in terms of number of foreground objects (1–4) and texture (synthetic images with simple shapes and simple to natural images with complex texture and cluttered background). These datasets include the synthetic CLEVR$n$ [21], SYNTH-CAR$n$ and SYNTH-CHAIR$n$, and the real REAL-CAR [54], where $n$ is the number of foreground objects. Additional details and results are included in the supplementary material.

**Implementation details.**    We assume a fixed and known number of objects of the same type. Fore- and background generators have similar architectures and the same number of output channels, but foreground generators have twice as many channels in the learnt constant tensor. Since foreground objects are smaller than the background, we set scale=1 for the background object, and randomly sample scales $<1$ for foreground objects. Please see our supplemental material for more implementation details and an ablation experiment. We make our code publicly available at github.com/thunguyenphuoc/BlockGAN.

**4.1. Qualitative results.**    Despite being trained with only unlabelled images, Figure 4 shows that BlockGAN learns to disentangle different objects within a scene: foreground from background, and between multiple foreground objects. More importantly, BlockGAN also provides explicit control and enables smooth manipulation of each object's pose $\boldsymbol{\theta}_i$ and identity $\mathbf{z}_i$. Figure 6 shows results on natural images with a cluttered background, where BlockGAN is still able to separate objects and enables 3D object-centric modifications. Since BlockGAN combines deep object features into scene features, changes in an object's properties also influence its shadows, and highlights adapt to the object's movement. These effects can be better observed in the supplementary animations.

**4.2. Quantitative results.**    We evaluate the visual fidelity of BlockGAN's results using Kernel Inception Distance [KID; 4], which has an unbiased estimator and works even for a small number of images. Note that KID does *not* measure the quality of object disentanglement, which is the main contribution of BlockGAN. We first compare with a vanilla GAN [WGAN-GP; 15] using a publicly

## One foreground object

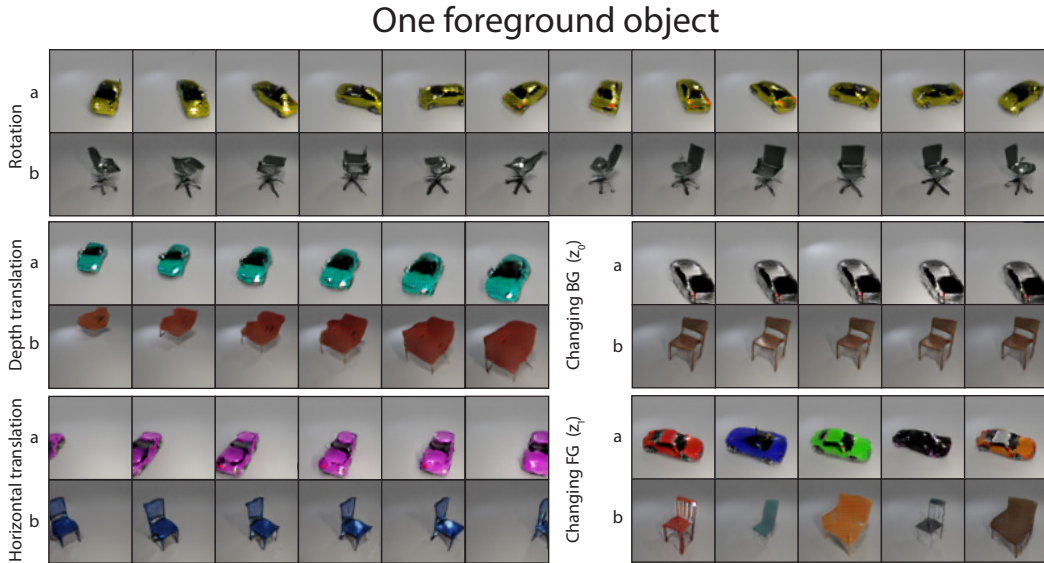

## Multiple foreground objects

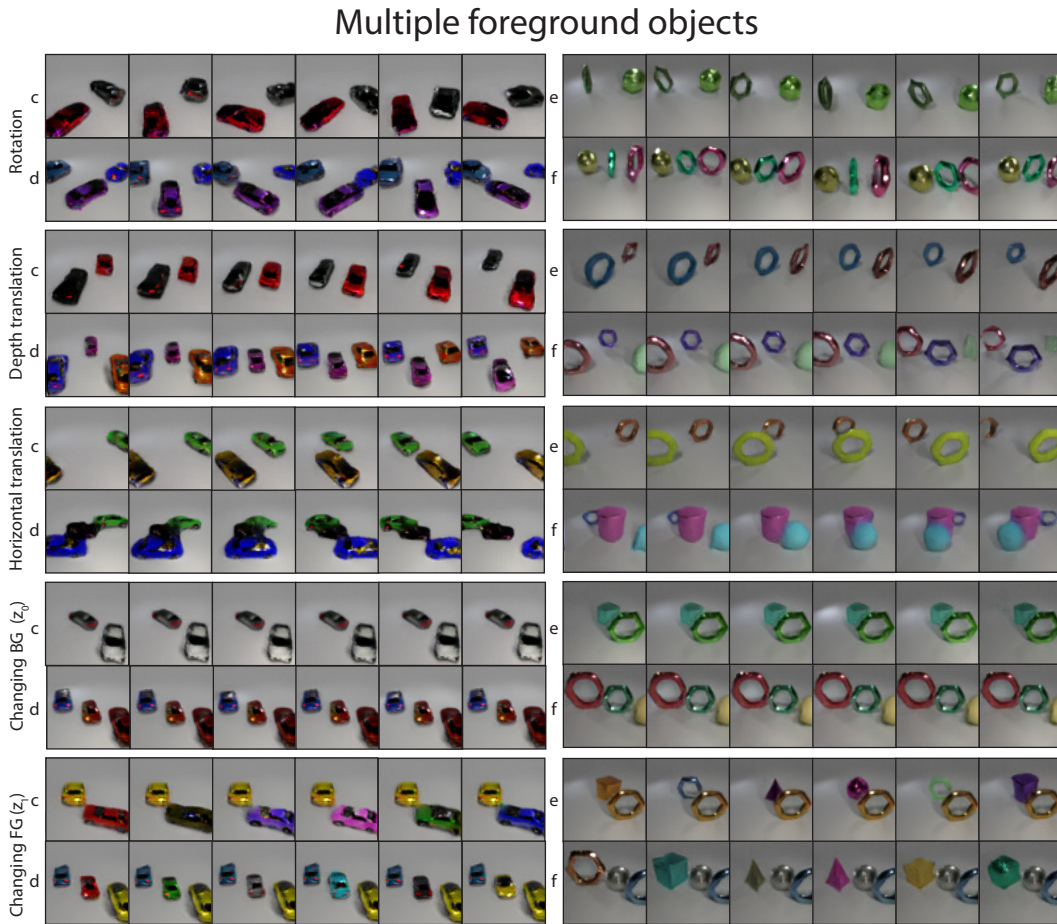

Figure 4: BlockGAN enables explicit spatial manipulation of individual objects (rotation, translation) and changing the identity of background or foreground objects across different datasets: (a) SYNTH-CAR1, (b) SYNTH-CHAIR1, (c) SYNTH-CAR2, (d) SYNTH-CAR3, (e) CLEVR2 and (f) CLEVR3. Notice how the shadows and highlights change as objects move around in the scene, and how changing the background lighting affects the appearance of foreground objects. Figure 6 shows similar results on natural images. Please refer to the supplemental material for animated results.

Table 1: KID estimates (mean ± std), lower is better, between real images and images generated by BlockGAN and other GANs. BlockGAN achieves competitive KID scores while providing control of each object in the generated images (which is not measured by KID).

| Method | SYNTH-CAR1 64×64 | SYNTH-CHAIR1 64×64 | REAL-CAR 64×64 | CLEVR2 64×64 |
|---|---|---|---|---|
| WGAN-GP [15] | 0.141 ± 0.002 | 0.111 ± 0.002 | 0.035 ± 0.001 | 0.076 ± 0.002 |
| LR-GAN [53] | **0.038 ± 0.001** | 0.036 ± 0.002 | **0.014 ± 0.001** | 0.052 ± 0.001 |
| HoloGAN [38] | 0.070 ± 0.001 | 0.058 ± 0.002 | 0.028 ± 0.002 | 0.032 ± 0.001 |
| BlockGAN (ours) | 0.039 ± 0.001 | **0.031 ± 0.001** | 0.016 ± 0.001 | **0.021 ± 0.001** |

available implementation[1] Secondly, we compare with LR-GAN [53], a 2D-based method that learns to generate image background and foregrounds separately and recursively. Finally, we compare with HoloGAN, which learns 3D scene representations that separate camera pose and identity, but does not consider object disentanglement. For LR-GAN and HoloGAN, we use the authors' code. We tune hyperparameters and then compute the KID for 10,000 images generated by each model (samples by all methods are included in the supplementary material). Table 1 shows that BlockGAN generates images with competitive or better visual fidelity than other methods.

**4.3. Scene manipulation beyond training data.** We show that at test time, 3D object features learnt by BlockGAN can be realistically manipulated in ways that have not been observed during training time. First, we show that the learnt 3D object features can also be reused to add more objects to the scene at test time, thanks to the compositionality inductive bias and our choice of scene composer function. Firstly, we use BlockGAN trained on datasets with only *one* foreground object and *one* background, and show that more foreground objects of the same category can be added to the same scene at *test time*. Figure 6 shows that 2–4 new objects are added and manipulated just like the original objects while maintaining realistic shadows and highlights. In Figure 5, we use BlockGAN trained on CLEVR4 and then remove (top) and add (bottom) more objects to the scene. Note how BlockGAN generates realistic shadows and occlusion for scenes that the model has never seen before.

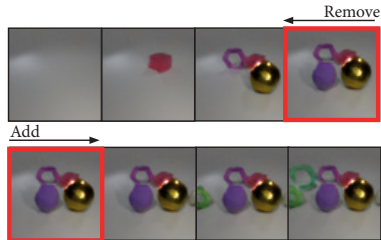

Figure 5: Removing/adding objects. The red box shows the original scene.

Secondly, we apply spatial manipulations that were not part of the similarity transform used during training, such as horizontal stretching, or slicing and combining different foreground objects. Figure 6 shows that object features can be geometrically modified intuitively, without needing explicit 3D geometry or multi-view supervision during training.

**4.4. Comparison to 2D-based LR-GAN.** LR-GAN [53] first generates a 2D background layer, and then generates and combines foreground layers with the generated background using alpha-compositing. Both BlockGAN and LR-GAN show the importance of combining objects in a contextually relevant manner to generate visually realistic images (see Table 1). However, LR-GAN does not offer explicit control over object location. More importantly, LR-GAN learns an *entangled* representation of the scene: sampling a different background noise vector also changes the foreground (Figure 7). Finally, unlike BlockGAN, LR-GAN does not allow adding more foreground objects during test time. This demonstrates the benefits of learning disentangled *3D* object features compared to a 2D-based approach.

**4.5. Ablation study: Non-uniform pose distribution.** For the natural REAL-CAR dataset, we observe that BlockGAN has difficulties learning the full 360° rotation of the car, even though fore- and background are disentangled well. We hypothesise that this is due to the mismatch between the true (unknown) pose distribution of the car, and the uniform pose distribution we assume during training. To test this, we create a synthetic dataset similar to SYNTH-CAR1 with a limited range of rotation, and train BlockGAN with a *uniform* pose distribution. To generate the imbalanced rotation dataset, we sample the rotation uniformly from the front/left/back/right viewing directions ±15°. In other words, the car is

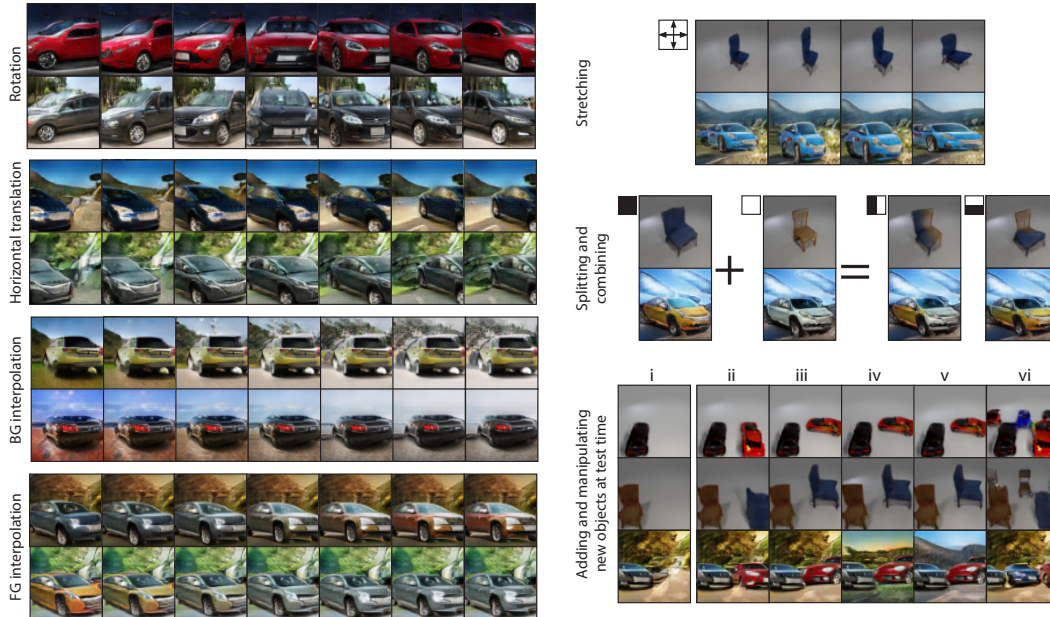

Figure 6: **Left:** REAL-CAR. Even for natural images with cluttered backgrounds, BlockGAN can still disentangle objects in a scene well. Note that interpolating the background ($z_0$) affects the appearance of the car in a meaningful way, showing the benefit of 3D scene features. **Right:** Test-time geometric modification of the learnt 3D object features (unless stated, background is fixed): stretching (top), splitting and combining (middle), and adding and manipulating new objects after training (bottom). Bottom row shows (i) Original scene, (ii) new object added, (iii) manipulated, (iv,v) different background appearance, and (vi) more objects added. Note the realistic lighting and shadows.

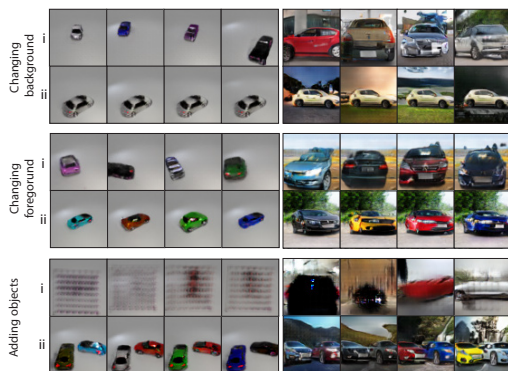

Figure 7: Comparison between (i) LR-GAN [53] and (ii) BlockGAN for SYNTH-CAR1 (left) and REAL-CAR (right).

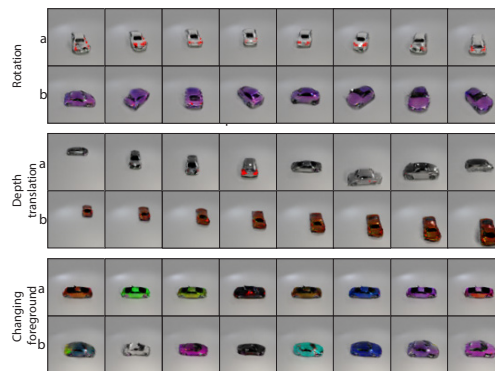

Figure 8: Different manipulations applied to BlockGAN trained on (a) a dataset with imbalanced rotations, and (b) a balanced dataset.

only seen from the front/left/back/right 30°, respectively, and there are four evenly spaced gaps of 60° that are never observed, for example views from the front-right. With the imbalanced dataset, Figure 8 (bottom) shows correct disentangling of foreground and background. However, rotation of the car only produces images with (near-)frontal views (top), while depth translation results in cars that are randomly rotated sideways (middle). We observe similar behaviour for the natural REAL-CAR dataset. This suggests that learning object disentanglement and full 3D pose rotation might be two independent problems. While assuming a uniform pose distribution already enables good object disentanglement, learning the pose distribution from the training data would likely improve the quality of 3D transforms.

In our supplemental material, we include comparisons to HoloGAN [38] as well as additional ablation studies on comparing different scene composer functions, using a perspective camera versus a weak-

perspective camera, adopting the style discriminator for scenes with cluttered backgrounds, and training on images with an incorrect number of objects.

# 5 Discussion and Future Work

We introduced BlockGAN, an image generative model that learns 3D object-aware scene representations from unlabelled images. We show that BlockGAN can learn a disentangled scene representation both in terms of objects and their properties, which allows geometric manipulations not observed during training. Most excitingly, even when BlockGAN is trained with fewer or even single objects, additional 3D object features can be added to the scene features at test time to create novel scenes with multiple objects. In addition to computer graphics applications, this opens up exciting possibilities, such as combining BlockGAN with models like BiGAN [10] or ALI [11] to learn powerful object representations for scene understanding and reasoning.

Future work can adopt more powerful relational learning models [25, 45, 51] to learn more complex object interactions such as inter-object shadowing or reflections. Currently, we assume prior knowledge of object category and the number of objects for training. We also assume object poses are uniformly distributed and independent from each other. Therefore, the ability to learn this information directly from training images would allow BlockGAN to be applied to more complex datasets with a varying number of objects and different object categories, such as COCO [30] or LSUN [57].

## Acknowledgments and Disclosure of Funding

We received support from the European Union's Horizon 2020 research and innovation programme under the Marie Skłodowska-Curie grant agreement No. 665992, the EPSRC Centre for Doctoral Training in Digital Entertainment (EP/L016540/1), RCUK grant CAMERA (EP/M023281/1), an EPSRC-UKRI Innovation Fellowship (EP/S001050/1), and an NVIDIA Corporation GPU Grant. We received a gift from Adobe.

## Broader Impact

BlockGAN is an image generative model that learns an object-oriented 3D scene representation directly from unlabelled 2D images. Our approach is a new machine learning technique that makes it possible to generate unseen images from a noise vector, with unprecedented control over the identity and pose of multiple independent objects as well as the background. In the long term, our approach could enable powerful tools for digital artists that facilitate artistic control over realistic procedurally generated digital content. However, any tool can in principle be abused, for example by adding new, manipulating or removing existing objects or people from images.

At training time, our network performs a task somewhat akin to scene understanding, as our approach learns to disentangle between multiple objects and individual object properties (specifically their pose and identity). At test time, our approach enables sampling new images with control over pose and identity for each object in the scene, but does not directly take any image input. However, it is possible to embed images into the latent space of generative models [1]. A highly realistic generative image model and a good image fit would then make it possible to approximate the input image and, more importantly, to edit the individual objects in a pictured scene. Similar to existing image editing software, this enables the creation of image manipulations that could be used for ill-intended misinformation (*fake news*), but also for a wide range of creative and other positive applications. We expect the benefits of positive applications to clearly outweigh the potential downsides of malicious applications.

## Footnotes

[1] https://github.com/LynnHo/DCGAN-LSGAN-WGAN-WGAN-GP-Tensorflow

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
