[Supplementary Material 1 · BlockGAN-Supplemental-Document.pdf]

# BlockGAN: Learning 3D Object-aware Scene Representations from Unlabelled Images
# — Supplemental Document —

**Thu Nguyen-Phuoc**
University of Bath

**Christian Richardt**
University of Bath

**Long Mai**
Adobe Research

**Yong-Liang Yang**
University of Bath

**Niloy Mitra**
Adobe Research & UCL

In this document, we show additional results (Section 1), ablation studies (Section 2) and comparisons (Section 3). We also provide additional information about our losses (Section 4), datasets (Section 5) and implementation, including our training procedure and network architectures (Section 6).

## 1 Additional results

Figure 1: Samples from HoloGAN [6] trained on the datasets (a) SYNTH-CHAIR1, (b) SYNTH-CAR2 and (c) CLEVR2. HoloGAN tends to associate each pose $\theta$ with a fixed object's identity, i.e., moving objects erroneously changes identity of both foreground and background (see top left, bottom left and right), while changing the noise vector $\mathbf{z}$ only changes a small part of the background (top right).

### 1.1 Comparison to *entangled* 3D scene representation

We compare BlockGAN with HoloGAN [6], which also learns deep 3D scene features but does not consider object disentanglement. In particular, HoloGAN only considers one noise vector $\mathbf{z}$ for identity and one pose $\theta$ for the entire scene, and does not consider translation $\mathbf{t}$ as part of $\theta$. While HoloGAN works well with object-centred scenes, it struggles with moving foreground objects. Figure 1 shows that HoloGAN tends to associate each pose $\theta$ with a fixed object's identity (i.e., moving objects erroneously changes identity of both foreground and background), while changing $\mathbf{z}$

only changes a small part of the background. BlockGAN, on the other hand, can separate identity and pose for *each* object, while being able to learn scene-level effects such as lighting and shadows.

## 1.2 Increasing the number of foreground objects

To test the capability of our method, we train BlockGAN on CLEVR6 (6 foreground objects). As shown in Figure 2, BlockGAN is still capable of generating and manipulating 3D object features, although the background generator now also produces foreground objects (Figure 2f). Moreover, rotating individual object leads to changes in object's depth (Figure 2a).

Interestingly, we notice that BlockGAN now generates images with more or less than 6 objects, despite being trained with images that contain exactly 6 objects (Figure 2g). We hypothesise that

Figure 2: Qualitative results of BlockGAN trained on CLEVR6. (a) Rotating a foreground object, (b) Horizontal translation, (c) Depth translation, (d) Changing foreground object 1, (e) Changing foreground object 3, (f) Changing the background object, and (g) Random samples.

BlockGAN's failure in this case is due to our assumption that the poses of all objects are independent from each other (during training, we randomly sample the pose $\theta$ for each object). This is not true in the physical world (also in the CLEVR dataset) where objects do not intersect. The more objects there are in the scene, the stronger the interdependence between objects' poses becomes. Therefore, for future work, we hope to adopt more powerful relation learning structures to learn objects' pose directly from training images. Another interesting direction is to design object-aware discriminators, which are capable of recognising fake images when the generators produce samples with more objects than the training images.

## 2 Additional ablation studies

### 2.1 Learning without the perspective camera

Here we show the advantage of implementing the perspective camera explicitly, compared to using a weak-perspective projection like HoloGAN [6]. Since a perspective camera directly affects foreshortening, it provides strong cues for BlockGAN to solve the scale/depth ambiguity. This is especially important for BlockGAN to learn to project and reason over occlusion by concatenating the depth and channel dimension, followed by an MLP. Since the MLP is flexible, BlockGAN trained *without* a perspective camera, therefore, tends to learn to associate an object's identity with scale and depth, while changing depth only changes the object's appearance (see Figure 3).

Figure 3: The effect of modelling the perspective camera explicitly (b) compared to using a weak-perspective camera (a). Note that with the weak-perspective camera (a), translation along the depth dimension (top) leads to identity changes without any translation in depth, while changing the noise vector $\mathbf{z}_1$ (bottom) changes both depth translation and, to a lesser extent, the object identity. Using a perspective camera correctly disentangles position and identity (b).

### 2.2 Scene composer function

We consider and compare three scene composer functions: (i) element-wise summation, (ii) element-wise maximum, and (iii) an MLP (multi-layer perceptron). We train BlockGAN with each function and compare their performance in terms of visual quality (KID score) in Table 1. While all three functions can successfully combine objects into a scene, the element-wise maximum performs best and easily generalises to multiple objects. Therefore, we use the element-wise maximum for BlockGAN.

### 2.3 Learning without the style discriminator

When BlockGAN is trained with a standard discriminator on datasets with a cluttered background, such as the REAL-CAR dataset, the foreground object features tend to include part of the background object. This creates visual artefacts when objects move in the scene (indicated by red arrows in Figure 4a). We hypothesise that these artefacts should be picked up by the discriminator since

Table 1: KID estimates for different scene composer functions.

| Method | SYNTH-CAR1 (64×64) | SYNTH-CHAIR1 (64×64) |
|--------|--------------------|-----------------------|
| Sum | $0.040 \pm 0.002$ | $0.038 \pm 0.001$ |
| MLP | $0.044 \pm 0.001$ | $0.033 \pm 0.001$ |
| Max | $\mathbf{0.039 \pm 0.002}$ | $\mathbf{0.031 \pm 0.001}$ |

generated images should look unrealistic. Therefore, we add more powerful style discriminators [6] to the original discriminator at different layers (see Section 4 for details). Figure 4b shows that the generator is indeed discouraged from adding background information to the foreground object features, leading to cleaner results.

Figure 4: With a standard discriminator (a), a part of the background appearance is baked into the foreground object (see red arrows). Adding the style discriminator (b) cleanly separates the car from the background.

## 2.4 Incorrect number of objects

We next investigate the performance of BlockGAN when the training data contains fewer or more objects than expected. In Figure 5, we show BlockGAN configured with 2 foreground object generators when trained with images containing 1 or 3 foreground objects. If only a single object is present (Figure 5, left), changing either of the two foreground generators changes the object's appearance and pose (top), while changing the background works as expected (bottom). If there are three objects present (Figure 5, right), changing one foreground generator changes one object as expected (top), while changing the background generator simultaneously changes one foreground object and the background (bottom).

Figure 5: Results for BlockGAN trained with 2 foreground (FG) object generators when trained on 1 or 3 foreground objects. 1 object (left): Changing either FG object changes the object's appearance and pose; changing the background works as expected. 3 objects (right): Changing one FG object changes one object as expected; changing the background changes one FG object and the background.

## 3 Comparison to other methods

In Figure 6, 7, 8 and 9, we show generated samples by a vanilla GAN (WGAN-GP [3]), 2D object-aware LR-GAN [7], 3D-aware HoloGAN [6] and our BlockGAN. Compared to other models, BlockGAN produces samples with competitive or better quality, *and* offers explicit control over the poses of objects in the generated images. Notice that although LR-GAN is designed to handle

Figure 6: Samples from WGAN-GP, LR-GAN, HoloGAN and our BlockGAN trained on SYNTH-CAR1.

foreground and background objects explicitly, for CLEVR2 with two foreground objects, this method struggles and tends to always place one foreground object at the image centre (see Figure 8).

**Implementation details** For WGAN-GP, we use a publicly available implementation[1]. For LR-GAN and HoloGAN, we use the code provided by the authors. We conduct hyperparameter search for these models, and report best results for each method. Note that for HoloGAN, we modify the 3D transformation to add translation during training, since this method assumes that foreground objects are at the image centre.

Figure 7: Samples from WGAN-GP, LR-GAN, HoloGAN and our BlockGAN trained on SYNTH-CHAIR1.

Figure 8: Samples from WGAN-GP, LR-GAN, HoloGAN and our BlockGAN trained on CLEVR2.

WGAN-GP

LR-GAN

HoloGAN

BlockGAN (ours)

Figure 9: Samples from WGAN-GP, LR-GAN, HoloGAN and our BlockGAN trained on REAL-CARS.

# 4 Loss function and style discriminator

For datasets with cluttered backgrounds like the natural REAL-CAR dataset, we adopt *style discriminators* in addition to the normal image discriminator (see the benefit in Figure 4). Style discriminators perform the same real/fake classification task as the standard image discriminator, but at the feature level across different layers. In particular, style discriminators classify the mean $\boldsymbol{\mu}$ and standard deviation $\boldsymbol{\sigma}$ of the features $\boldsymbol{\Phi}_l$ at different levels $l$ (which are believed to describe the image "style"). The mean $\boldsymbol{\mu}(\boldsymbol{\Phi}_l(\mathbf{x}))$ and variance $\boldsymbol{\sigma}(\boldsymbol{\Phi}_l(\mathbf{x}))$ of the features $\boldsymbol{\Phi}_l(\mathbf{x})$ are computed across batch and spatial dimensions independently using:

$$\boldsymbol{\mu}(\boldsymbol{\Phi}_l(\mathbf{x})) = \frac{1}{N \times H \times W} \sum_{n=1}^{N} \sum_{h=1}^{H} \sum_{w=1}^{W} \boldsymbol{\Phi}_l(\mathbf{x})_{nhw}, \tag{1}$$

$$\boldsymbol{\sigma}(\boldsymbol{\Phi}_l(\mathbf{x})) = \sqrt{\frac{1}{N \times H \times W} \sum_{n=1}^{N} \sum_{h=1}^{H} \sum_{w=1}^{W} \left(\boldsymbol{\Phi}_l(\mathbf{x})_{nhw} - \boldsymbol{\mu}(\boldsymbol{\Phi}_l(\mathbf{x}))\right)^2 + \epsilon}. \tag{2}$$

The style discriminators are implemented as MLPs with sigmoid activation functions for binary classification. A style discriminator at layer $l$ is written as

$$L_{\text{style}}^{l}(\mathbf{G}) = \mathbb{E}_{\mathbf{z},\theta}[-\log \ \mathrm{D}_l(\mathbf{G}(\mathbf{z},\theta))]. \tag{3}$$

The total loss therefore can be written as

$$L_{\text{total}}(\mathbf{G}) = L_{\text{GAN}}(\mathbf{G}) + \lambda_{\text{s}} \cdot \sum_{l} L_{\text{style}}^{l}(\mathbf{G}). \tag{4}$$

We set $\lambda_{\text{s}} = 1$ for all natural datasets and $\lambda_{\text{s}} = 0$ for synthetic datasets.

# 5 Datasets

We modify the CLEVR dataset [4] to add a larger variety of colours and primitive shapes. Additionally, we use the scene setups provided by CLEVR to render the remaining synthetic datasets (SYNTH-CAR$n$ and SYNTH-CHAIR$n$, with $n$ foreground objects each). These include a fixed, grey background, a virtual camera with fixed parameters but random location jittering, and random lighting. We also use the render script from CLEVR to randomly place foreground objects into the scene and render them. We render all image at resolution $128 \times 128$, and bi-linearly downsample them to $64 \times 64$ for training. For the natural CAR dataset, each image is first scaled such that the smaller side is 64, then it is cropped to produce a $64 \times 64$ pixel crop. During training, we randomly move the $64 \times 64$ cropping window before cropping the image. Figure 10 includes samples from our generated datasets, and Table 2 lists the range of pose parameters used for each dataset during training.

Link for 3D textured chair models:
https://keunhong.com/publications/photoshape/

Link for CLEVR:
https://github.com/facebookresearch/clevr-dataset-gen

Link for natural CAR dataset:
http://mmlab.ie.cuhk.edu.hk/datasets/comp_cars/

Table 2: Datasets used in our paper ($n$ = number of foreground objects). 'Azimuth' describes object rotation about the up-axis. 'Elevation' refers to the camera's elevation above ground. 'Scaling' is the scale factor applied to foreground objects. 'Horiz. transl.' and 'Depth transl.' are horizontal/depth translation of objects relative to the global origin. Ranges represent uniform random distributions.

| Name | # Images | Azimuth | Elevation | Scaling | Horiz. transl. | Depth transl. |
|------|----------|---------|-----------|---------|----------------|---------------|
| SYNTH-CAR$n$ | 80,000 | $0° - 359°$ | $45°$ | $0.5 - 0.6$ | $-5 - 5$ | $-5 - 5$ |
| SYNTH-CHAIR$n$ | 100,000 | $0° - 359°$ | $45°$ | $0.5 - 0.6$ | $-5 - 5$ | $-5 - 5$ |
| CLEVR$n$ [4] | 100,000 | $0° - 359°$ | $45°$ | $0.5 - 0.6$ | $-4 - 4$ | $-4 - 4$ |
| REAL-CARS [8] | 139,714 | $0° - 359°$ | $0° - 35°$ | $0.5 - 0.8$ | $-3 - 4$ | $-5 - 6$ |

Figure 10: Samples from the synthetic datasets.

# 6 Implementation

## 6.1 Training details

**Virtual camera model.** We assume a virtual camera with a focal length of 35 mm and a sensor size of 32 mm (Blender's default values), which corresponds to an angle of view of $2 \arctan \frac{32\,\text{mm}}{2 \times 35\,\text{mm}} = 49.1$ degrees (we use the same setup for natural images).

**Sampling.** We initialise all weights using $\mathcal{N}(0, 0.2)$ and biases as 0. For CLEVR$n$, we use noise vector dimensions of $|\mathbf{z}_0| = 20$ for the background, and $|\mathbf{z}_i| = 60$ (for $i = 1, \ldots, n$) for the foreground objects, to account for their relative visual complexity. Similarly, for SYNTH-CAR$n$ and SYNTH-CHAIR$n$, we use $|\mathbf{z}_0| = 30$ and $|\mathbf{z}_i| = 90$ (for $i = 1, \ldots, n$), to account for their relative visual complexity. For the natural REAL-CAR dataset, we use $|\mathbf{z}_0| = 100$ and $|\mathbf{z}_1| = 200$. Note that we only feed $\mathbf{z}$ to the 3D features of each object, and not to the 3D scene features and 2D features. Table 2 provides the ranges we use for sampling the pose $\boldsymbol{\theta}_i$ of foreground objects during training.

**Training.** We train BlockGAN using the Adam optimiser [5], with $\beta_1 = 0.5$ and $\beta_2 = 0.999$. We use the same learning rate for both the discriminator and the generator. Empirically, we find that updating the generator twice for every update of the discriminator achieves images with the best

visual fidelity. We use a learning rate of 0.0001 for all synthetic datasets. For the natural CARS dataset, we use a learning rate of 0.00005. We train all datasets with a batch size of 64 for 50 epochs. Training takes 1.5 days for the synthetic datasets and 3 days for the natural REAL-CARS dataset.

**Infrastructure.** All models were trained using a single GeForce RTX 2080 GPU.

## 6.2 Network architecture

We describe the network architecture for the BlockGAN foreground object generator in Table 3, the BlockGAN background generator in Table 4, and the overall BlockGAN generator in Tables 5 and 6 for synthetic and real datasets, respectively. Note that we use ReLU for the synthetic datasets and LReLU for the natural CAR dataset after the AdaIN layer. The discriminator is described in Table 7.

In terms of the notation in Section 3 of the main paper, object features have dimensions $H_o \times W_o \times D_o \times C_o = 16 \times 16 \times 16 \times 64$, scene features have the same dimensions $H_s \times W_s \times D_s \times C_s = 16 \times 16 \times 16 \times 64$, and camera features have dimensions $H_c \times W_c = 16 \times 16$ (before up-convolutions to $64 \times 64$) with $C_c = 64$ channels for synthetic datasets and $C_c = 256$ channels for natural image datasets.

As GANs empirically tend to perform better on category-specific datasets, we decided to start with this assumption. A promising future direction is to adopt a shared rendering layer for objects generated by different category-specific generators, similar to Aliev et al. [1].

Table 3: Network architecture of the BlockGAN foreground (FG) object generator.

| Layer type | Kernel size | Stride | Normalisation | Output dimension |
|---|---|---|---|---|
| Learnt constant tensor | — | — | AdaIN | $4 \times 4 \times 4 \times 512$ |
| UpConv | $3 \times 3 \times 3$ | 2 | AdaIN | $8 \times 8 \times 8 \times 128$ |
| UpConv | $3 \times 3 \times 3$ | 2 | AdaIN | $16 \times 16 \times 16 \times 64$ |
| 3D transformation | — | — | — | $16 \times 16 \times 16 \times 64$ |

Table 4: Network architecture of the BlockGAN background (BG) object generator.

| Layer type | Kernel size | Stride | Normalisation | Output dimension |
|---|---|---|---|---|
| Learnt constant tensor | — | — | AdaIN | $4 \times 4 \times 4 \times 256$ |
| UpConv | $3 \times 3 \times 3$ | 2 | AdaIN | $8 \times 8 \times 8 \times 128$ |
| UpConv | $3 \times 3 \times 3$ | 2 | AdaIN | $16 \times 16 \times 16 \times 64$ |
| 3D transformation | — | — | — | $16 \times 16 \times 16 \times 64$ |

Table 5: Network architecture of the BlockGAN generator for all synthetic datasets.

| Layer type | Kernel size | Stride | Activation | Norm. | Output dimension |
|---|---|---|---|---|---|
| $n\times$FG generator (Table 3) | — | — | ReLU | — | $16 \times 16 \times 16 \times 64$ |
| BG generator (Table 4) | — | — | ReLU | — | $16 \times 16 \times 16 \times 64$ |
| Element-wise maximum | — | — | — | — | $16 \times 16 \times 16 \times 64$ |
| Concatenate | — | — | — | — | $16 \times 16 \times (16 \cdot 64)$ |
| Conv | $1 \times 1$ | 1 | ReLU | — | $16 \times 16 \times 64$ |
| UpConv | $4 \times 4$ | 2 | ReLU | AdaIN | $32 \times 32 \times 64$ |
| UpConv | $4 \times 4$ | 2 | ReLU | AdaIN | $64 \times 64 \times 64$ |
| UpConv | $4 \times 4$ | 1 | ReLU | AdaIN | $64 \times 64 \times 3$ |

Table 6: Network architecture of the BlockGAN generator for the REAL-CARS dataset. Differences to the synthetic foreground object generator in Table 5 are highlighted in blue.

| Layer type | Kernel size | Stride | Activation | Normal. | Output dimension |
|---|---|---|---|---|---|
| FG generator (Table 3) | — | — | LReLU | — | $16 \times 16 \times 16 \times 64$ |
| BG generator (Table 4) | — | — | LReLU | — | $16 \times 16 \times 16 \times 64$ |
| Element-wise maximum | — | — | — | — | $16 \times 16 \times 16 \times 64$ |
| Concatenate | — | — | — | — | $16 \times 16 \times (16 \cdot 64)$ |
| Conv | $1 \times 1$ | 1 | LReLU | — | $16 \times 16 \times 256$ |
| UpConv | $4 \times 4$ | 2 | LReLU | AdaIN | $32 \times 32 \times 128$ |
| UpConv | $4 \times 4$ | 2 | LReLU | AdaIN | $64 \times 64 \times 64$ |
| UpConv | $4 \times 4$ | 1 | LReLU | AdaIN | $64 \times 64 \times 3$ |

Table 7: Network architecture of the BlockGAN discriminator for both synthetic and real datasets.

| Layer type | Kernel size | Stride | Activation | Normalisation | Output dimension |
|---|---|---|---|---|---|
| Conv | $5 \times 5$ | 2 | LReLU | IN/Spectral | $32 \times 32 \times 64$ |
| Conv | $5 \times 5$ | 2 | LReLU | IN/Spectral | $16 \times 16 \times 128$ |
| Conv | $5 \times 5$ | 2 | LReLU | IN/Spectral | $8 \times 8 \times 256$ |
| Conv | $5 \times 5$ | 2 | LReLU | IN/Spectral | $4 \times 4 \times 512$ |
| Fully connected | — | — | Sigmoid | None/Spectral | 1 |

## Footnotes

[1]https://github.com/LynnHo/DCGAN-LSGAN-WGAN-WGAN-GP-Tensorflow

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

[Supplementary Material 2]

# BlockGAN: Learning 3D Object-aware Scene Representations from Unlabelled Images — Supplemental Document —

**Thu Nguyen-Phuoc**
University of Bath

**Christian Richardt**
University of Bath

**Long Mai**
Adobe Research

**Yong-Liang Yang**
University of Bath

**Niloy Mitra**
Adobe Research & UCL

In this document, we show additional results (Section 1), ablation studies (Section 2) and comparisons (Section 3). We also provide additional information about our losses (Section 4), datasets (Section 5) and implementation, including our training procedure and network architectures (Section 6).

## 1   Additional results

Figure 1: Samples from HoloGAN [6] trained on the datasets (a) SYNTH-CHAIR1, (b) SYNTH-CAR2 and (c) CLEVR2. HoloGAN tends to associate each pose $\theta$ with a fixed object's identity, i.e., moving objects erroneously changes identity of both foreground and background (see top left, bottom left and right), while changing the noise vector **z** only changes a small part of the background (top right).

### 1.1   Comparison to *entangled* 3D scene representation

We compare BlockGAN with HoloGAN [6], which also learns deep 3D scene features but does not consider object disentanglement. In particular, HoloGAN only considers one noise vector **z** for identity and one pose $\theta$ for the entire scene, and does not consider translation **t** as part of $\theta$. While HoloGAN works well with object-centred scenes, it struggles with moving foreground objects. Figure 1 shows that HoloGAN tends to associate each pose $\theta$ with a fixed object's identity (i.e., moving objects erroneously changes identity of both foreground and background), while changing **z**

only changes a small part of the background. BlockGAN, on the other hand, can separate identity and pose for *each* object, while being able to learn scene-level effects such as lighting and shadows.

## 1.2 Increasing the number of foreground objects

To test the capability of our method, we train BlockGAN on CLEVR6 (6 foreground objects). As shown in Figure 2, BlockGAN is still capable of generating and manipulating 3D object features, although the background generator now also produces foreground objects (Figure 2f). Moreover, rotating individual object leads to changes in object's depth (Figure 2a).

Interestingly, we notice that BlockGAN now generates images with more or less than 6 objects, despite being trained with images that contain exactly 6 objects (Figure 2g). We hypothesise that

Figure 2: Qualitative results of BlockGAN trained on CLEVR6. (a) Rotating a foreground object, (b) Horizontal translation, (c) Depth translation, (d) Changing foreground object 1, (e) Changing foreground object 3, (f) Changing the background object, and (g) Random samples.

BlockGAN's failure in this case is due to our assumption that the poses of all objects are independent from each other (during training, we randomly sample the pose $\theta$ for each object). This is not true in the physical world (also in the CLEVR dataset) where objects do not intersect. The more objects there are in the scene, the stronger the interdependence between objects' poses becomes. Therefore, for future work, we hope to adopt more powerful relation learning structures to learn objects' pose directly from training images. Another interesting direction is to design object-aware discriminators, which are capable of recognising fake images when the generators produce samples with more objects than the training images.

## 2 Additional ablation studies

### 2.1 Learning without the perspective camera

Here we show the advantage of implementing the perspective camera explicitly, compared to using a weak-perspective projection like HoloGAN [6]. Since a perspective camera directly affects foreshortening, it provides strong cues for BlockGAN to solve the scale/depth ambiguity. This is especially important for BlockGAN to learn to project and reason over occlusion by concatenating the depth and channel dimension, followed by an MLP. Since the MLP is flexible, BlockGAN trained *without* a perspective camera, therefore, tends to learn to associate an object's identity with scale and depth, while changing depth only changes the object's appearance (see Figure 3).

Figure 3: The effect of modelling the perspective camera explicitly (b) compared to using a weak-perspective camera (a). Note that with the weak-perspective camera (a), translation along the depth dimension (top) leads to identity changes without any translation in depth, while changing the noise vector $z_1$ (bottom) changes both depth translation and, to a lesser extent, the object identity. Using a perspective camera correctly disentangles position and identity (b).

### 2.2 Scene composer function

We consider and compare three scene composer functions: (i) element-wise summation, (ii) element-wise maximum, and (iii) an MLP (multi-layer perceptron). We train BlockGAN with each function and compare their performance in terms of visual quality (KID score) in Table 1. While all three functions can successfully combine objects into a scene, the element-wise maximum performs best and easily generalises to multiple objects. Therefore, we use the element-wise maximum for BlockGAN.

### 2.3 Learning without the style discriminator

When BlockGAN is trained with a standard discriminator on datasets with a cluttered background, such as the REAL-CAR dataset, the foreground object features tend to include part of the background object. This creates visual artefacts when objects move in the scene (indicated by red arrows in Figure 4a). We hypothesise that these artefacts should be picked up by the discriminator since

Table 1: KID estimates for different scene composer functions.

| Method | SYNTH-CAR1 (64×64) | SYNTH-CHAIR1 (64×64) |
|---|---|---|
| Sum | $0.040 \pm 0.002$ | $0.038 \pm 0.001$ |
| MLP | $0.044 \pm 0.001$ | $0.033 \pm 0.001$ |
| Max | $\mathbf{0.039 \pm 0.002}$ | $\mathbf{0.031 \pm 0.001}$ |

generated images should look unrealistic. Therefore, we add more powerful style discriminators [6] to the original discriminator at different layers (see Section 4 for details). Figure 4b shows that the generator is indeed discouraged from adding background information to the foreground object features, leading to cleaner results.

Figure 4: With a standard discriminator (a), a part of the background appearance is baked into the foreground object (see red arrows). Adding the style discriminator (b) cleanly separates the car from the background.

## 2.4 Incorrect number of objects

We next investigate the performance of BlockGAN when the training data contains fewer or more objects than expected. In Figure 5, we show BlockGAN configured with 2 foreground object generators when trained with images containing 1 or 3 foreground objects. If only a single object is present (Figure 5, left), changing either of the two foreground generators changes the object's appearance and pose (top), while changing the background works as expected (bottom). If there are three objects present (Figure 5, right), changing one foreground generator changes one object as expected (top), while changing the background generator simultaneously changes one foreground object and the background (bottom).

Figure 5: Results for BlockGAN trained with 2 foreground (FG) object generators when trained on 1 or 3 foreground objects. 1 object (left): Changing either FG object changes the object's appearance and pose; changing the background works as expected. 3 objects (right): Changing one FG object changes one object as expected; changing the background changes one FG object and the background.

## 3 Comparison to other methods

In Figure 6, 7, 8 and 9, we show generated samples by a vanilla GAN (WGAN-GP [3]), 2D object-aware LR-GAN [7], 3D-aware HoloGAN [6] and our BlockGAN. Compared to other models, BlockGAN produces samples with competitive or better quality, *and* offers explicit control over the poses of objects in the generated images. Notice that although LR-GAN is designed to handle

Figure 6: Samples from WGAN-GP, LR-GAN, HoloGAN and our BlockGAN trained on SYNTH-CAR1.

foreground and background objects explicitly, for CLEVR2 with two foreground objects, this method struggles and tends to always place one foreground object at the image centre (see Figure 8).

**Implementation details** For WGAN-GP, we use a publicly available implementation[1]. For LR-GAN and HoloGAN, we use the code provided by the authors. We conduct hyperparameter search for these models, and report best results for each method. Note that for HoloGAN, we modify the 3D transformation to add translation during training, since this method assumes that foreground objects are at the image centre.

Figure 7: Samples from WGAN-GP, LR-GAN, HoloGAN and our BlockGAN trained on SYNTH-CHAIR1.

WGAN-GP

LR-GAN

HoloGAN

BlockGAN (ours)

Figure 8: Samples from WGAN-GP, LR-GAN, HoloGAN and our BlockGAN trained on CLEVR2.

WGAN-GP

LR-GAN

HoloGAN

BlockGAN (ours)

Figure 9: Samples from WGAN-GP, LR-GAN, HoloGAN and our BlockGAN trained on REAL-CARS.

# 4 Loss function and style discriminator

For datasets with cluttered backgrounds like the natural REAL-CAR dataset, we adopt *style discriminators* in addition to the normal image discriminator (see the benefit in Figure 4). Style discriminators perform the same real/fake classification task as the standard image discriminator, but at the feature level across different layers. In particular, style discriminators classify the mean $\boldsymbol{\mu}$ and standard deviation $\boldsymbol{\sigma}$ of the features $\boldsymbol{\Phi}_l$ at different levels $l$ (which are believed to describe the image "style"). The mean $\boldsymbol{\mu}(\boldsymbol{\Phi}_l(\mathbf{x}))$ and variance $\boldsymbol{\sigma}(\boldsymbol{\Phi}_l(\mathbf{x}))$ of the features $\boldsymbol{\Phi}_l(\mathbf{x})$ are computed across batch and spatial dimensions independently using:

$$\boldsymbol{\mu}(\boldsymbol{\Phi}_l(\mathbf{x})) = \frac{1}{N \times H \times W} \sum_{n=1}^{N} \sum_{h=1}^{H} \sum_{w=1}^{W} \boldsymbol{\Phi}_l(\mathbf{x})_{nhw}, \tag{1}$$

$$\boldsymbol{\sigma}(\boldsymbol{\Phi}_l(\mathbf{x})) = \sqrt{\frac{1}{N \times H \times W} \sum_{n=1}^{N} \sum_{h=1}^{H} \sum_{w=1}^{W} \left(\boldsymbol{\Phi}_l(\mathbf{x})_{nhw} - \boldsymbol{\mu}(\boldsymbol{\Phi}_l(\mathbf{x}))\right)^2 + \epsilon}. \tag{2}$$

The style discriminators are implemented as MLPs with sigmoid activation functions for binary classification. A style discriminator at layer $l$ is written as

$$L_{\text{style}}^l(\mathbf{G}) = \mathbb{E}_{\mathbf{z},\theta}[-\log \ \mathbf{D}_l(\mathbf{G}(\mathbf{z},\theta))]. \tag{3}$$

The total loss therefore can be written as

$$L_{\text{total}}(\mathbf{G}) = L_{\text{GAN}}(\mathbf{G}) + \lambda_{\text{s}} \cdot \sum_l L_{\text{style}}^l(\mathbf{G}). \tag{4}$$

We set $\lambda_{\text{s}} = 1$ for all natural datasets and $\lambda_{\text{s}} = 0$ for synthetic datasets.

# 5 Datasets

We modify the CLEVR dataset [4] to add a larger variety of colours and primitive shapes. Additionally, we use the scene setups provided by CLEVR to render the remaining synthetic datasets (SYNTH-CAR$n$ and SYNTH-CHAIR$n$, with $n$ foreground objects each). These include a fixed, grey background, a virtual camera with fixed parameters but random location jittering, and random lighting. We also use the render script from CLEVR to randomly place foreground objects into the scene and render them. We render all image at resolution $128 \times 128$, and bi-linearly downsample them to $64 \times 64$ for training. For the natural CAR dataset, each image is first scaled such that the smaller side is 64, then it is cropped to produce a $64 \times 64$ pixel crop. During training, we randomly move the $64 \times 64$ cropping window before cropping the image. Figure 10 includes samples from our generated datasets, and Table 2 lists the range of pose parameters used for each dataset during training.

Link for 3D textured chair models:
https://keunhong.com/publications/photoshape/

Link for CLEVR:
https://github.com/facebookresearch/clevr-dataset-gen

Link for natural CAR dataset:
http://mmlab.ie.cuhk.edu.hk/datasets/comp_cars/

Table 2: Datasets used in our paper ($n$ = number of foreground objects). 'Azimuth' describes object rotation about the up-axis. 'Elevation' refers to the camera's elevation above ground. 'Scaling' is the scale factor applied to foreground objects. 'Horiz. transl.' and 'Depth transl.' are horizontal/depth translation of objects relative to the global origin. Ranges represent uniform random distributions.

| Name | # Images | Azimuth | Elevation | Scaling | Horiz. transl. | Depth transl. |
|---|---|---|---|---|---|---|
| SYNTH-CAR$n$ | 80,000 | $0° - 359°$ | $45°$ | $0.5 - 0.6$ | $-5 - 5$ | $-5 - 5$ |
| SYNTH-CHAIR$n$ | 100,000 | $0° - 359°$ | $45°$ | $0.5 - 0.6$ | $-5 - 5$ | $-5 - 5$ |
| CLEVR$n$ [4] | 100,000 | $0° - 359°$ | $45°$ | $0.5 - 0.6$ | $-4 - 4$ | $-4 - 4$ |
| REAL-CARS [8] | 139,714 | $0° - 359°$ | $0° - 35°$ | $0.5 - 0.8$ | $-3 - 4$ | $-5 - 6$ |

Figure 10: Samples from the synthetic datasets.

# 6 Implementation

## 6.1 Training details

**Virtual camera model.** We assume a virtual camera with a focal length of 35 mm and a sensor size of 32 mm (Blender's default values), which corresponds to an angle of view of $2 \arctan \frac{32\,\text{mm}}{2 \times 35\,\text{mm}} = 49.1$ degrees (we use the same setup for natural images).

**Sampling.** We initialise all weights using $\mathcal{N}(0, 0.2)$ and biases as 0. For CLEVR$n$, we use noise vector dimensions of $|\mathbf{z}_0| = 20$ for the background, and $|\mathbf{z}_i| = 60$ (for $i = 1, \ldots, n$) for the foreground objects, to account for their relative visual complexity. Similarly, for SYNTH-CAR$n$ and SYNTH-CHAIR$n$, we use $|\mathbf{z}_0| = 30$ and $|\mathbf{z}_i| = 90$ (for $i = 1, \ldots, n$), to account for their relative visual complexity. For the natural REAL-CAR dataset, we use $|\mathbf{z}_0| = 100$ and $|\mathbf{z}_1| = 200$. Note that we only feed $\mathbf{z}$ to the 3D features of each object, and not to the 3D scene features and 2D features. Table 2 provides the ranges we use for sampling the pose $\boldsymbol{\theta}_i$ of foreground objects during training.

**Training.** We train BlockGAN using the Adam optimiser [5], with $\beta_1 = 0.5$ and $\beta_2 = 0.999$. We use the same learning rate for both the discriminator and the generator. Empirically, we find that updating the generator twice for every update of the discriminator achieves images with the best

visual fidelity. We use a learning rate of 0.0001 for all synthetic datasets. For the natural CARS dataset, we use a learning rate of 0.00005. We train all datasets with a batch size of 64 for 50 epochs. Training takes 1.5 days for the synthetic datasets and 3 days for the natural REAL-CARS dataset.

**Infrastructure.** All models were trained using a single GeForce RTX 2080 GPU.

## 6.2 Network architecture

We describe the network architecture for the BlockGAN foreground object generator in Table 3, the BlockGAN background generator in Table 4, and the overall BlockGAN generator in Tables 5 and 6 for synthetic and real datasets, respectively. Note that we use ReLU for the synthetic datasets and LReLU for the natural CAR dataset after the AdaIN layer. The discriminator is described in Table 7.

In terms of the notation in Section 3 of the main paper, object features have dimensions $H_o \times W_o \times D_o \times C_o = 16 \times 16 \times 16 \times 64$, scene features have the same dimensions $H_s \times W_s \times D_s \times C_s = 16 \times 16 \times 16 \times 64$, and camera features have dimensions $H_c \times W_c = 16 \times 16$ (before up-convolutions to $64 \times 64$) with $C_c = 64$ channels for synthetic datasets and $C_c = 256$ channels for natural image datasets.

As GANs empirically tend to perform better on category-specific datasets, we decided to start with this assumption. A promising future direction is to adopt a shared rendering layer for objects generated by different category-specific generators, similar to Aliev et al. [1].

Table 3: Network architecture of the BlockGAN foreground (FG) object generator.

| Layer type | Kernel size | Stride | Normalisation | Output dimension |
| --- | --- | --- | --- | --- |
| Learnt constant tensor | — | — | AdaIN | $4 \times 4 \times 4 \times 512$ |
| UpConv | $3 \times 3 \times 3$ | 2 | AdaIN | $8 \times 8 \times 8 \times 128$ |
| UpConv | $3 \times 3 \times 3$ | 2 | AdaIN | $16 \times 16 \times 16 \times 64$ |
| 3D transformation | — | — | — | $16 \times 16 \times 16 \times 64$ |

Table 4: Network architecture of the BlockGAN background (BG) object generator.

| Layer type | Kernel size | Stride | Normalisation | Output dimension |
| --- | --- | --- | --- | --- |
| Learnt constant tensor | — | — | AdaIN | $4 \times 4 \times 4 \times 256$ |
| UpConv | $3 \times 3 \times 3$ | 2 | AdaIN | $8 \times 8 \times 8 \times 128$ |
| UpConv | $3 \times 3 \times 3$ | 2 | AdaIN | $16 \times 16 \times 16 \times 64$ |
| 3D transformation | — | — | — | $16 \times 16 \times 16 \times 64$ |

Table 5: Network architecture of the BlockGAN generator for all synthetic datasets.

| Layer type | Kernel size | Stride | Activation | Norm. | Output dimension |
|---|---|---|---|---|---|
| $n\times$FG generator (Table 3) | — | — | ReLU | — | $16 \times 16 \times 16 \times 64$ |
| BG generator (Table 4) | — | — | ReLU | — | $16 \times 16 \times 16 \times 64$ |
| Element-wise maximum | — | — | — | — | $16 \times 16 \times 16 \times 64$ |
| Concatenate | — | — | — | — | $16 \times 16 \times (16 \cdot 64)$ |
| Conv | $1 \times 1$ | 1 | ReLU | — | $16 \times 16 \times 64$ |
| UpConv | $4 \times 4$ | 2 | ReLU | AdaIN | $32 \times 32 \times 64$ |
| UpConv | $4 \times 4$ | 2 | ReLU | AdaIN | $64 \times 64 \times 64$ |
| UpConv | $4 \times 4$ | 1 | ReLU | AdaIN | $64 \times 64 \times 3$ |

Table 6: Network architecture of the BlockGAN generator for the REAL-CARS dataset. Differences to the synthetic foreground object generator in Table 5 are highlighted in blue.

| Layer type | Kernel size | Stride | Activation | Normal. | Output dimension |
|---|---|---|---|---|---|
| FG generator (Table 3) | — | — | LReLU | — | $16 \times 16 \times 16 \times 64$ |
| BG generator (Table 4) | — | — | LReLU | — | $16 \times 16 \times 16 \times 64$ |
| Element-wise maximum | — | — | — | — | $16 \times 16 \times 16 \times 64$ |
| Concatenate | — | — | — | — | $16 \times 16 \times (16 \cdot 64)$ |
| Conv | $1 \times 1$ | 1 | LReLU | — | $16 \times 16 \times 256$ |
| UpConv | $4 \times 4$ | 2 | LReLU | AdaIN | $32 \times 32 \times 128$ |
| UpConv | $4 \times 4$ | 2 | LReLU | AdaIN | $64 \times 64 \times 64$ |
| UpConv | $4 \times 4$ | 1 | LReLU | AdaIN | $64 \times 64 \times 3$ |

Table 7: Network architecture of the BlockGAN discriminator for both synthetic and real datasets.

| Layer type | Kernel size | Stride | Activation | Normalisation | Output dimension |
|---|---|---|---|---|---|
| Conv | $5 \times 5$ | 2 | LReLU | IN/Spectral | $32 \times 32 \times 64$ |
| Conv | $5 \times 5$ | 2 | LReLU | IN/Spectral | $16 \times 16 \times 128$ |
| Conv | $5 \times 5$ | 2 | LReLU | IN/Spectral | $8 \times 8 \times 256$ |
| Conv | $5 \times 5$ | 2 | LReLU | IN/Spectral | $4 \times 4 \times 512$ |
| Fully connected | — | — | Sigmoid | None/Spectral | 1 |

## Footnotes

[1]https://github.com/LynnHo/DCGAN-LSGAN-WGAN-WGAN-GP-Tensorflow