[Reviews · NeurIPS 2020]

Review 1

Summary and Contributions: [Main Contributions] 1- The proposed method allows generating 3D scene representations flexibly in a compositional fashion. Significance: very high 2- The proposed approach is a good step in the direction of modeling the computer graphics pipeline (creating geometry, manipulating them and doing rendering) using neural networks. Significance: high 3- The proposed method learns such 3D representations without any supervision and only requires 2D RGB images during training. Significance: high Reviewer’s legend for the level of significance: [very low, low, medium, high, very high] The ratings are a function of the results/analysis shown in the paper, current state of the art methods and the reviewer’s perspective on the authors’ approach [High-level summary of the paper] This paper proposes a generative model that operates on compositional, implicit 3D scene representations and is trained using low resolution (64 x 64) 2D images only without any additional supervision signal. The model is trained on scenes with 1-4 objects but, thanks to compositionality, it can generate scenes with more number of objects. [Low-level summary] The model first generates {1,…,K} foreground object(s) and a background separately using two decoder neural networks that map a noise vector, Z, whose elements control object appearance properties and some transformation parameters (rotation, translation, scaling) to “3D features”. The 3D features are then composed through taking their element-wise maximum and then concatenation to form a “unified 3D scene representation”. For rendering, the unified scene representation is fed to another neural network (the renderer) which also takes as input the camera parameters and produces a rendering of the scene. The model is trained with a GAN-based objective. The authors evaluate their model on multiple datasets such as CLEVR, Real-Car and Synthetic cars and chairs. The main experiments they conduct is as follow 1) To show that their model has learned disentangled representations (i.e. they change attributes of the foreground objects or the background and show the renderings which reflect those changes) 2) To evaluate visual fidelity, they show that their model achieves nearly the same or lower KID estimates compared to other methods for all datasets 3) They train a model on scenes with 1 object but show that they can use that model to generate scenes up to 5 objects of the same category with different attributes 4) They show that they can manipulate some of the attributes (e.g. stretching) that were not seen during training and get renderings that reflect such changes to a degree.

Strengths: - The proposed method in this work is novel (specially the compositionality part) and the authors have conducted good experiments to evaluate their model and make their case. - The paper’s organization is good and each section is well-developed overall (introduction may need a bit of work). - The results clearly show that the authors’ proposed approach works well compared to prior work Technical strengths: - Out of distribution generalization: Due to compositionality, the model trained on a 1-4 objects can generate scenes with higher number of objects - No supervision except using unlabeled 2D images - Given the difficulty of the problem, the renderer seems to be doing a good job - The authors show that their model can deal with basic forms of occlusion

Weaknesses: - Despite that the contributions of the work are good but the results do not meet the expectations set by paper’s tone and the claims of the authors make (see below for more details on this) - The fact that the object generator is category-specific makes application of the proposed approach limited Technical weaknesses: - The proposed model does not not have/use an inference mechanism - There is no guarantee that the elements of Z are interpretable or correspond to a meaningful disentangled 3D feature - The background is plain white for all of the experiments - The authors mention in the methods section that the object generator is category-specific but it is possible that the renderer is also category-specific because the authors do not say whether or not this is the case this in the text and do not provide any results on that either.

Correctness: [Claims] - No supervision on pose, class labels, object masks multiple views geometry: correct, except for pose because the pose is given as input - Maintaining image realism: partially/mostly correct. The renderer does a good job overall (e.g. handling reflections, casting shadows properly most of the time) but details even for symmetric shapes such as cars are easily missed. This could be due to issues with the object representations too. - Interactions of object appearance: partially correct - Reasoning over occlusion: Partially correct. Although the proposed approach can deal with occlusion to some degree but it cannot do a very good job. For instance when the torus-like CLEVR object is in the front and the object in the back moves, the moving object is not visible through torus’ open area. - Learning disentangled representations: partially correct; changing pose or properties of most objects results in slight changes in other appearance/transformation/visual properties of the same objects too. This happens much less for CLEVR, which is a simpler dataset. For all datasets, only sampling new backgrounds seems to leave almost every other attribute of the objects completely intact. - Able to generate scenes with arbitrary number of objects: not correct. Adding more than 2 or more objects usually alters the way other objects look like in the scene as if they have a new identity. In the supplementary videos the authors also mainly show this capability for the Real Cars dataset. - Visual fidelity of objects in the scene: not correct. This is only mostly correct for the CLEVR and sometimes the chairs datasets but for the other datasets only the texture and overall formation of object gives clues that the object should be a car. Additionally, the background sometimes behaves as foreground.

Clarity: The paper is written well and the authors have done a good job to developed each section. The supplementary materials contains additional results; thanks to the authors for this. The authors may however work a bit on the introduction to improve it based on the comments I gave earlier.

Relation to Prior Work: The authors have done a good job to review prior works and describe their shortcomings in addition to how their work has tried to address them. I included two references that I think the authors should cite.

Reproducibility: Yes

Additional Feedback: [Additional Comments] Abstract: - Lines 4-5: This sentence can be written better → Meanwhile, work that considers scene compositionality treats scene objects only as image patches or 2D layers with alpha maps Introduction: - The introduction is good overall but it would be better if the authors revise it so that it looks more realistic with respect to what the authors have shown in the paper. The beginning paragraphs of the introduction might raise the hope/expectation of readers that the authors’ method will somehow address the mentioned shortcomings of the prior works in an ideal fashion. Related Work: - Lines 67-68: It is not clear what the authors mean by the limited success of GAN-based models when applied to natural images with respect of their work. I have seen plenty of GAN-based models that generate realistic natural scene images - Lines 70-72: It would be better if the authors can be more explicit about the inductive bias. At the beginning they talk about compositionality as an inductive bias but do they mean other things as well? For instance, using transposed 3D convolutions (which I believe is not important) or using object transformation parameters or scene camera information? - Line 74: The work of Sitzmann et al. (reference 46) does not use CNNs. Replace CNNs by neural networks Method: - The method section is well-developed and the supplementary materials provide further details/results in addition to implementation details. - Figure 1 could be a bit misleading unless the reader reads parts of the paper. The 3D features might not be as interpretable as shown in the figure unless the authors can show that the 3D feature maps can be readily interpreted or visualized as 3D objects. - Lines 132-133: The introduction would be a better place for the last sentence here - In figure 5 right: top row is stretching and middle row is splitting and combining Typos: - Unlabelled → Unlabeled [ Missing References] - Dosovitskiy, Alexey, et al. "Learning to generate chairs, tables and cars with convolutional networks." IEEE transactions on pattern analysis and machine intelligence 39.4 (2016): 692-705. - Zhou, Tinghui, et al. "View synthesis by appearance flow." European conference on computer vision. Springer, Cham, 2016. [Rebuttal Requests] Rebuttal (prioritized): - I am curious to see how adding 1-6 objects for a model trained on Synthetic Cars or CLEVR datasets while changing their transformation properties affects the original object(s) in the scene. Also, assuming that a model has been trained on scenes with 4 objects, I am also curious to see what happens if the objects are removed from the scene during test time. A video for both of these would be helpful for me. - Is the renderer model also category-specific? If not, I would like to see some renderings of scenes with composition of synthetic cars and other objects (e.g. chairs). - Resolution of the output images is low which makes it hard to judge the quality of the results sometimes. Have the authors tried to train their model to produce images with higher resolution? - To me the “background” is mostly about scene lighting. Why didn’t the authors try changing the background color or texture during training? - The authors used transposed 3D convolutional operators to obtain object and background features. Is there a reason for that? Note that using transposed 3D convolutional operators does not make the proposed method more similar to how humans perceive the world necessarily. I wonder, did the authors also try using transposed 2D convolutional operators to see if they can get similar results? - Why the authors have not trained their model on using shapes of some of the categories in the ShapeNet Core dataset?


Review 2

Summary and Contributions: This paper presents a generative image model that disentangles objects/background appearance, orientation, and viewpoint. The proposed method samples noise vectors for the background and a set of objects, which are oriented in space and then mapped to 3D voxel grids with deep features, combined into a single 3D volume, and rendered. At test time, this enables effects when sampling the generative model such as rendering views with different numbers of objects, rotating the objects, and moving the camera viewpoint. After rebuttal discussion comments: I appreciate the authors' rebuttal experiment that helped satisfy my curiosity regarding a mismatched number of expected vs. actual objects. The rebuttal did not really address my concerns regarding the rendering method, but I still stand by the paper's strengths and still vote that the paper should be accepted.

Strengths: The overall method and strategy seem sound, and the results are quite impressive. I am generally strongly in favor of this research direction of baking in more 3D graphics knowledge into deep learning models (including generative image models in the case of this paper), and I think that this paper is a good point of evidence that supports this overall trend.

Weaknesses: The approach is limited to low-resolution images of relatively simple scenes where we know the number of objects. What happens if an incorrect number of foreground objects are generated/used? Additionally, some of the specific method details regarding composing and rendering could be better justified. In 3.2, the scene composition is described as an elementwise maximum of the features for each object at each location. Wouldn't this result in a composed feature vector at a 3D location being a mix of different dimensions of different objects? Or is elementwise maximum actually supposed to be a max-pooling over just the object dimension and not the feature dimension? Why not something more straightforward like a linear combination of features where one of the feature dimensions is treated as the weight? Some of the details regarding the scene composition and learned rendering seem quite similar to the DeepVoxels paper, and should be discussed. I am also curious why the projection/depth composition is learned. It seems like this would require the network to learn correct occlusion (which appeared to be an issue in the results of the DeepVoxels paper by causing slight viewpoint inconsistencies, and later works such as Neural Volumes and NeRF remedied this issue by using explicit opacity/density instead of learning the occlusion composition). Maybe this choice of rendering function should be discussed more.

Correctness: Yes, the paper's claims, algorithm, and experiments seem correct.

Clarity: Yes, the paper is overall well-written and easy to understand. One high-level comment is that much of the paper is written describing improvements/changes relative to HoloGAN [36], which makes the paper feel a bit incremental.

Relation to Prior Work: Yes, the discussion of prior work seems reasonable.

Reproducibility: Yes

Additional Feedback: Line 2-3: maybe say "scene generative modeling" instead of "scene representation learning", since representation learning is more general and many prior works in that space do not have the issues mentioned in this sentence. Line 23 and elsewhere: it looks awkward to include the abbreviation within the square brackets for the reference. Line 87: learned --> have been proposed for learning Section 3.3 A lot of the details in the second paragraph are common knowledge in computer vision/3D geometry (perspective projection, coordinate transformations, sampling along perspective rays, etc.), and maybe do not need to be included in the paper. Line 180: This paragraph's descriptions of the transformations as matrix multiplications are also pretty common and seem more like implementation details.


Review 3

Summary and Contributions: The authors propose a generative model that represents a scene using 3D representations of the scenes constituent objects.

Strengths: The paper is well motivated, the method is explained well and the visual results are clear, informative and demonstrated on a variety of datasets. It is interesting to decompose the object into pose and identity features and leads to some really nice visualisations in Figure 4 showing how the model deals well with translation and rotation. Results in Figure 6 are also very nice, showing that object can be added to the scene.

Weaknesses: A number of implementation details are missing from the main body of the paper but they authors do provide code (though I have not tried to run it).

Correctness: The authors show strong results that substitute each of their three claims.

Clarity: The paper is well written.

Relation to Prior Work: Yes, this work improves on previous work by showing that objects can be added and manipulated.

Reproducibility: Yes

Additional Feedback: Ln 135-136 TYPO: which helps to disentanglement between pose and identity


Review 4

Summary and Contributions: This paper propose BlockGAN, a GAN variant that trains on unlabeled images and learns to explicitly model objects with 3D features and compose them to render, using a graphics-inspired neural pipeline.

Strengths: Incorporating 3D representations and object compositionality into GANs has been an important direction. This paper contributes to this direction by bringing together several components: voxel-like 3D object features disentangled from pose, scene composer as element-wise maximum over object features, and scene rendering by perspective projection. These components are not brand new ideas individually, but they follow graphics insight and are put together in a novel way to model object geometry and composition without explicit supervision.

Weaknesses: Weaknesses: - The voxel-like 3D object features are low-resolution, and seem hard to scale due to their cubic size. This paper only trains on 64x64 images. - The voxel-like 3D object features also lack geometry-texture disentanglement, which limits its modeling capacity for objects with more complex geometry and texture. For example, cars in Figure 5 Left do not look totally realistic despite their low resolution. * After rebuttal: some of my above concerns about representation choices are relieved. - I also have concerns for the practicability of the claimed "manipulation", as it can neither control the source image to operate on (e.g. given a user image, as opposed to randomly sampled one), nor fully control the manipulation direction (e.g. user wants the car to be green, as opposed to a random color). As pointed out in "Broader Impact", learning image encoding along with the generative network could be a fix, but seems hard.

Correctness: The method is reasonable and graphics-inspired. The experiments are standard.

Clarity: I basically understand the method section, but more math and formulas could be added. Some symbols go unexplained, like H_o, H_c, H_s. The experiment section is informative.

Relation to Prior Work: - The work claims "Current work on scene representation learning either ignores scene background or treats the whole scene as one object. Meanwhile, work that considers scene compositionality treats scene objects only as image patches or 2D layers with alpha maps." But there is prior work that combines object compositionality and 3D awareness in generative modeling, and perform 3D-aware image manipulation, like [1]. - It might also be valuable to compare different 3D representation choices (like [2], [3], [4]) in more detail, and discuss which can be used to compose multiple objects. This can be a plus for model design justification. [1] 3D-Aware Scene Manipulation via Inverse Graphics. NeurIPS 2018. [2] Scene Representation Networks: Continuous 3D-Structure-Aware Neural Scene Representations. NeurIPS 2019. [3] Visual object networks: Image generation with disentangled 3D representations. NeurIPS 2018. [4] HoloGAN: Unsupervised learning of 3D representations from natural images. ICCV 2019. * After rebuttal: I appreciate authors' address, and I recognize the setup differences. It would be nice to adjust claims and related work accordingly.

Reproducibility: Yes

Additional Feedback:

[Author Response · NeurIPS 2020]

We thank the reviewers for their encouraging feedback. We will revise our writing as suggested (R1, R2 and R4), and
discuss the prior work mentioned by the reviewers in the final manuscript (R1, R2 and R4).

BlockGAN is a generative model that learns 3D **object-aware** scene representations using only *unlabelled* images.
We show that BlockGAN works with both synthetic datasets with simple backgrounds and real images with complex
background and lighting. We evaluated BlockGAN on 64×64 images, which are common for this line of work [1,2].
Due to resource constraints, we did not train BlockGAN on images with higher resolution, but instead chose to focus on
pushing the complexity, both in terms of number of objects and, especially, in terms of texture and cluttered background,
of the datasets. We believe that this is the first demonstration that deep 3D **object** representations can be learnt directly
from natural images without any template geometry, pretrained object detector, or multi-view input.

**R1:** *Claims:* Although pose is an input to our model, no GT pose labels were used for training. Hence, we maintain
our claim that we do *not* need any pose *supervision*. In addition to synthetic images with a simple background, we
also train BlockGAN on the real CAR dataset with complex, natural backgrounds (see Figures 5 and 6). Changing the
background object, in this case, changes not only lighting but also colour and texture.
*Learnt renderer:* Yes, it is category-specific. We discuss a shared renderer as future work in line 131 of the supplement.
*Objects' appearance interaction:* As the foreground object moves, its appearance and shadow move accordingly,
depending on where the object is in relation to the camera view (specularity) and lighting positions (shadows) – this can
be observed most clearly in the animated results. Indeed, we leave more complex effects, such as inter-object reflection
between foreground objects, as future works as discussed in line 272 in the paper.

*3D vs 2D convolutions:* Our goal is to perform 3D transformations on deep 3D
features (including the background). Performing scene combination in 3D allows
representing geometry and appearance independent of camera specification.
*Removing objects:* We show adding and removing objects on the right →
*ShapeNet dataset:* ShapeNet contains only a limited number of textured models,
most of which are low quality, e.g., no specularity. Note that our SYNTH-CHAIR
dataset contains ShapeNet chairs with high-quality textures from PhotoShape.

**R2:** In the final version of the paper, we will also add a discussion on the projec-
tion/depth composition method, in addition to the rendering function. *Number of*
*objects:* On the right, we show BlockGAN with 2 foreground (FG) object gener-
ators trained with images containing 1 or 3 FG objects. **1 object (top):** Changing
either FG object changes the object's appearance and pose; changing the back-
ground works as expected. **3 objects (bottom):** Changing one FG object changes
one object as expected; changing the background changes one FG object and the
background. *Compositing function:* We perform max pooling across objects. This
does not require any learning, and, more importantly, is agnostic to the number of
inputs, allowing any number of objects to be added at test time. We have set up an
experiment with a learned linear weight per voxel, as suggested by R2; however,
the training collapsed, and we could not get it to work during the rebuttal period.

**R4:** *Performance of voxel grids:* Voxel grids can be more memory efficient when
adopting warping fields [Neural Volumes, SIGGRAPH 2019], a multi-resolution
strategy [Lighthouse, CVPR 2020] or sparsity [Neural Sparse Voxel Fields, arXiv

2020]. Moreover, HoloGAN [ICCV 2019] showed that voxel grids with low spatial resolution but high feature dimension
can be very expressive. Note that the choice of voxel grid does not affect whether shape and appearance can be separated
– both voxel grids [HoloGAN] and implicit functions [Texture Fields, CVPR 2020] can separate shape and appearance.
*Image encoding:* Many GAN models (apart from ALI and BiGAN) lack an inference (image encoding) mechanism.
However, recent work on training image encoders, such as Image2StyleGAN [ICCV 2019], exploit the representations
learnt by GANs to great effect. We look forward to extending BlockGAN towards this direction in future work.
*Alternative representations:* We thank the reviewer for pointing out [3], which we will cite and discuss. However, this
work requires predefined 3D mesh templates for each category (which are not always available) and a pretrained Mask
R-CNN to detect objects in images, while BlockGAN learns to disentangle and represent objects using only unlabelled
2D images. Scene Representation Networks [NeurIPS 2019] need many images with labelled poses to learn a good
representation for each scene. More importantly, only Visual Object Networks [NeurIPS 2018] and HoloGAN [ICCV
2019], which both use voxel grids, are trained successfully in an unsupervised manner, and can work across multiple
scenes. This makes voxel grids a reasonable and effective choice to achieve the goals of our paper.

*References:* [1] Investigating object compositionality in Generative Adversarial Networks. Neural Networks 2020.
[2] Towards Unsupervised Learning of Generative Models for 3D Controllable Image Synthesis, CVPR 2020. [3]
3D-Aware Scene Manipulation via Inverse Graphics. NeurIPS 2018.


[Meta-Review · NeurIPS 2020]

Four knowledgeable referees support accept and I accept. We encourage and expect the authors to incorporate the reviewers' suggestions for improving the paper. In particular, please show how the rendering changes by adding/removing 1-6 objects using a model trained on scenes with 4 objects, and please address R2's concerns regarding the rendering method. NOTE FROM PROGRAM CHAIRS: The paper is accepted, however please revise and expand the Broader Impact statement in the camera-ready version. The current statement is biased towards potential positive effects of the work and does not adequately address the risks of 'ill-intended image manipulation'.